# TorsionNet: A Reinforcement Learning Approach to Sequential Conformer Search

**Tarun Gogineni**[1], **Ziping Xu**[2], **Exequiel Punzalan**[3], **Runxuan Jiang**[1],
**Joshua Kammeraad**[2,3], **Ambuj Tewari**[2], **Paul Zimmerman**[3]

[1]Department of EECS, University of Michigan
[2]Department of Statistics, University of Michigan
[3]Department of Chemistry, University of Michigan
{tgog,zipingxu,epunzal,runxuanj,joshkamm,tewaria,paulzim}@umich.edu

## Abstract

Molecular geometry prediction of flexible molecules, or conformer search, is a long-standing challenge in computational chemistry. This task is of great importance for predicting structure-activity relationships for a wide variety of substances ranging from biomolecules to ubiquitous materials. Substantial computational resources are invested in Monte Carlo and Molecular Dynamics methods to generate diverse and representative conformer sets for medium to large molecules, which are yet intractable to chemoinformatic conformer search methods. We present TorsionNet, an efficient sequential conformer search technique based on reinforcement learning under the rigid rotor approximation. The model is trained via curriculum learning, whose theoretical benefit is explored in detail, to maximize a novel metric grounded in thermodynamics called the Gibbs Score. Our experimental results show that TorsionNet outperforms the highest scoring chemoinformatics method by 4x on large branched alkanes, and by several orders of magnitude on the previously unexplored biopolymer lignin, with applications in renewable energy. TorsionNet also outperforms the far more exhaustive but computationally intensive Self-Guided Molecular Dynamics sampling method.

## 1   Introduction

Accurate prediction of likely 3D geometries of flexible molecules is a long standing goal of computational chemistry, with broad implications for drug design, biopolymer research, and QSAR analysis. However, this is a very difficult problem due to the exponential growth of possible stable physical structures, or conformers, as a function of the size of a molecule. Levinthal's infamous paradox notes that a medium sized protein polypeptide chain exposes around $10^{143}$ possible torsion angle combinations, indicating brute force to be an intractable search method for all but the smallest molecules [21]. While the conformational space of a molecule's rotatable bonds is continuous with an infinite number of possible *conformations*, there are a finite number of stable, low energy *conformers* that lie in a local minimum on the energy surface [26]. Research in pharmaceuticals and bio-polymer material design can be accelerated by developing efficient methods for low energy conformer search of large molecules.

Take the example of *lignin*, a class of chemically complex branched biopolymer that has great potential as a renewable biofuel [32, 56]. The challenge in taking advantage of lignin is its structural complexity that makes it hard to selectively break down into useful chemical components [45]. Effective strategies to make use of lignin require deep understanding of its chemical reaction pathways, which in turn require accurate sampling of conformational behavior [4, 24]. Molecular dynamics (MD) simulations (though expensive) is the usual method for sampling complex molecules such as lignin [37, 54]. Understanding lignin processing on a molecular level using MD appears essential for improving their degradation efficiencies in mechano-chemical experimental processes [19].

**Conformer generation and rigid rotor model.** The goal of conformer generation is to build a representative set of conformers to "cover" the likely conformational space of a molecule, and sample its energy landscape well [8]. To that end, many methods have been employed [8, 15] to generate diverse sets of low energy conformers. Three notable cheminformatics methods are RDKit's Experimental-Torsion Distance Geometry with Basic Knowledge (ETKDG) [33], OpenBabel's Confab systematic search algorithm [30], and CORINA [39]. ETKDG and Confab are open source whereas CORINA is commercial. The latter focuses on generating a single low-energy conformer. ETKDG generates a distance bounds matrix to specify minimum and maximum distances each atomic pair in a molecule can take, and stochastically samples conformations that fit these bounds. On the other hand, Confab is a systematic search process, utilizing the *rigid rotor approximation* of fixing constant bond angles and bond lengths. With bond angles and lengths frozen, the only degrees of freedom for molecular geometry are the *torsion angles* of rotatable bonds, which Confab discretizes into buckets and then sequentially cycles through all combinations. It has been previously demonstrated that the exhaustive Confab search performs similarly to RDKit for molecules with small *rotatable bond number (rbn)*, but noticeably better for large, flexible ($rbn > 10$) molecules [8] if the compute time is available. Systematic search is intractable at very high $rbn$ ($> 50$) due to the combinatorial explosion of torsion angle combinations, whereas distance geometry fails entirely.

**Differences from protein folding.** Protein folding is a well-studied subproblem of conformer generation, where there is most often only one target conformer of a single, linear chain of amino acids. Protein folding is aided by vast biological datasets including structural homologies and genetic multiple sequence alignments (MSAs). In addition, the structural motifs for most finite sequences of amino acids are well known, greatly simplifying the folding problem. The few papers [2, 7, 16, 18] that apply machine learning methods to protein folding or conformer generation without any structural motifs predict only one target. *In contrast, the general conformer generation problem is a far broader challenge where the goal is to generate a set of representative conformers.* Additionally, there is insufficient database coverage for other complex polymers that are structurally different from proteins since they are not as immensely studied [15]. For these reasons, deep learning techniques such as Alphafold [41] developed for de novo protein generation do not have the same goal as we do.

**Main Contributions.** First, we argue that posing conformer search as a reinforcement learning problem has several benefits over alternative formulations including generative models. Second, we present TorsionNet, a conformer search technique based on Reinforcement Learning (RL). We make careful design choices in the use of MPNNs [12] with LSTMs [17] to generate independent torsion sampling distributions for all torsions at every timestep. Further, we construct a nonstationary reward function to model the task as a dynamic search process that conditions over histories. Third, we employ curriculum learning, a learning strategy that trains a model on simpler tasks and then gradually increases the task difficulty. In conformer search, we have a natural indication of task difficulty, namely the number of rotatable bonds, and size of the molecule. Fourth, we demonstrate that TorsionNet outperforms chemoinformatic methods in an environment of small and medium sized alkanes by up to 4x, and outclasses them by at least four orders of magnitude on a large lignin polymer. TorsionNet also performs around twice as well as the far more compute intensive Self-Guided MD (SGMD) on the lignin environment. We also demonstrate that TorsionNet has learned to detect important conformational regions. Curriculum learning is increasingly used in RL but we have little theoretical understanding for why it works [27]. Our final contribution is showing, via simple simple theoretical arguments, why curriculum learning might be able to reduce the sample complexity of simple exploration strategies in RL under suitable assumptions about task relatedness.

**Related work.** Recently there has been significant work using deep learning models for de novo drug target generation [53], property prediction [12], and conformer search [11, 23]. Some supervised approaches [23] require a target dataset of empirically measured molecule shapes, utilizing scarce data generated by expensive X-ray crystallography. Simm and Hernández-Lobato [43] utilize dense structural data of a limited class of small molecules generated from a computationally expensive MD simulation. To our knowledge, no previous works exist that attempt to find conformer sets of medium to large sized molecules. You et al. [53] and Wang et al. [48] utilize reinforcement learning on graph neural networks, but neither utilize recurrent units for memory nor action distributions constructed from subsets of node embeddings. Curriculum learning has been proposed as a way to handle non-convex optimization problems arising in deep learning [3, 34, 49]. There is empirical work showing that the RL training process benefits from a curriculum by starting with non-sparse reward signals, which mitigates the difficulties of exploration [1, 10, 28].

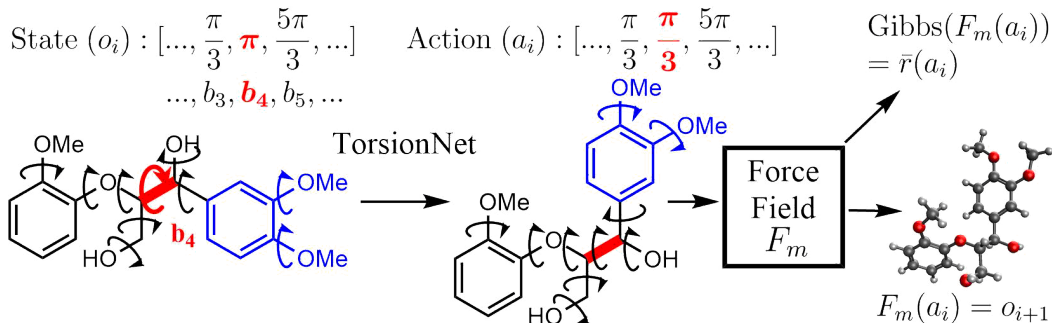

Figure 1: Conformer $o_i$ is the state defined by the molecule's torsion angles for each rotatable bond. TorsionNet receives conformer $o_i$ along with memory informed by previous conformers and outputs a set of new torsion angles $a_i$. The MMFF force field $\mathcal{F}_m$ then relaxes all atoms to local energy minimum $o_{i+1}$ and computes $\text{Gibbs}(o_{i+1}) = \bar{r}(a)$, the stationary reward.

## 2    Conformer Generation as a Reinforcement Learning Problem

We pose conformer search as an RL problem, which introduces several benefits over the generative models that individually place atoms in 3D space, or produce distance constraints. First and foremost, the latter models do not solve the problem of finding a *representative set of diverse, accessible conformers* since all conformations are generated in parallel without regard for repeats. Moreover, they require access to expensive empirical crystallographic or simulated MD data. Learning from physics alone is a long-standing goal in structure prediction challenges to reduce the need for expensive empirical data. To this end, we utilize a classical molecular force field approximation called MMFF [13] that can cheaply calculate the potential energy of conformational states and run gradient descent-based energy minimizations. Conformations that have undergone relaxation become conformers that lie stably at the bottom of a local potential well. RL-based conformer search is able to learn the conformational potential energy surface via the process depicted in Figure 1. RL is naturally adapted to the paradigm of sequential generation with the only training data being scalar energy evaluations as reward. Deep generative models [43] show reasonable performance for constructing geometries of molecules very similar to the training distribution, but their exploration ability is fundamentally limited by the ability to access expensive training sets.

We model the conformer generation problem as a contextual MDP [14, 25] with a non-stationary reward function, all possible molecular graph structures as the context space $\mathcal{X}$, the trajectory of searched conformers as the state space $\mathcal{S}$, the torsional space of a given molecule as the action space $\mathcal{A}$ and horizon $K$. This method can be seen as a deep augmentation of the Confab systematic search algorithm; instead of sequentially cycling through torsion combinations, we sample intelligently. As our goal is to find a set of good conformations, we use a non-stationary reward function, which encourages the agent to search for conformations that have not been seen during its history. Notably, our model learns from energy function and inter-conformer distance evaluations alone. We use a Message Passing Neural Network [12] as a feature extractor for the input graph structure to handle the exponentially large context space. We solve this large state and action space problem with the Proximal Policy Optimization (PPO) algorithm [36]. Finally, to improve the generalization ability of our training method, we apply a curriculum learning strategy [3], in which we train our model within a family of molecules in an imposed order. Next, we formally describe the problem setup.

### 2.1   Environment

**Context space.** Our context is the molecular graph structure, which is processed by a customized graph neural network, called TorsionNet. TorsionNet aggregates the structural information of a molecule efficiently for our RL problem. We will discuss TorsionNet in detail in the next subsection.

**Conformer space and state space.** The conformer space of a given molecule with $n$ independent torsions, or freely rotatable bonds, is defined by the torsional space $\mathcal{O} = [0, 2\pi]^n$. Since we optimize a non-stationary reward function, the agent requires knowledge of the entire sequence of searched conformers in order to avoid duplication. We compress the partially observed environment into an MDP by considering every sequence of searched conformers to be a unique state. This gives rise to the formalism $\mathcal{S} \subset \mathcal{O}^*$ and $s_t = (o_1, o_2, \ldots, o_t) \in \mathcal{O}^t$.

**Action space.** Our action space $\mathcal{A} \subset \mathcal{O}$ is the torsional space. Generating a conformer at each timestep can be modelled as simultaneously outputting torsion angles for each rotatable bond. We discretize the action space by breaking down each torsion angle $[0, 2\pi]$ into discrete angle buckets, i.e. $\{k\pi/3\}_{k=1}^6$. Each torsion angle is sampled independently of all the other torsions.

**Transition dynamics.** At each timestep, our model generates *unminimized* conformation $a_i \in \mathcal{A}$. Conformation $a_i$ then undergoes a first order optimization, using a molecular force field. We state that the minimizer $\mathcal{F}_m$ is a mapping $\mathcal{A} \mapsto \mathcal{O}$, which accepts input of output conformer $a_i$ and generates new *minimized* conformer for the next model step, as in $\mathcal{F}_m(a_i) = o_{i+1}$. Distinct generated conformations may minimize to the same or similar conformer.

**Gibbs Score.** To measure performance, we introduce a novel metric called the *Gibbs Score*, which has not directly been utilized in the conformer generation literature to date. Conformers of a molecule exist in nature as an interconverting equilibrium, with relative frequencies determined by a Gibbs distribution over energies. Therefore, the Gibbs score intends to measure the quality of a set of conformers with respect to a given force field function rather than distance to empirically measured conformations. It is designed as a *representativeness* measure of a finite conformation output set to the Gibbs partition function. For any $o \in \mathcal{O}$, we define Gibbs measure as

$$\text{Gibbs}(o) = \exp\left[-(E(o) - E_0)/k\tau\right]/Z_0,$$

where $E(o)$ is the corresponding energy of the conformation $o$, $k$ the Boltzmann constant, $\tau$ the thermodynamic temperature, and $Z_0$ and $E_0$ are normalizing scores and energies, respectively, for molecule $x$ gathered from a classical generation method as needed. The exponential function in the definition above can generate numerically unreliable rewards if the normalization factors $Z_0$ and $E_0$ are selected without consideration of the overall energy level. But they do not need to be set to their ground truth values for our method to be successful.

The Gibbs measure relates the energy of a conformer to its thermal accessibility at a specific temperature. The Gibbs score of a set $O$ is the sum of Gibbs measures for each unique conformer: $\text{Gibbs}(O) = \sum_{o \in O} \text{Gibbs}(o)$. With the Gibbs score, the total quality of the conformer set is evaluated, while guaranteeing a level of inter-conformer diversity with a distance measure that is described in the next paragraph. It can thereby be used to directly compare the quality of different output sets. Large values of this metric correspond to good coverage of the low-energy regions of the conformational space of a molecule. To our knowledge, this metric is the first one to attempt to examine both conformational diversity and quality at once.

**Horizons and rewards.** We train the model using a fixed episodic length $K$, which is chosen on a per environment basis based on number of torsions of the target molecule(s). We design the reward function to encourage a search for conformers with low energy and low similarity to minimized conformations seen during the current trajectory. We first describe the stationary reward function, which is the Gibbs measure after MMFF optimization:

$$\bar{r}(a) = Gibbs(\mathcal{F}_m(a)), \text{ for the proposed torsion angles } a \in \mathcal{A}.$$

To prune overly similar conformers, we create a nonstationary reward. For a threshold $m$, distance metric $d : \mathcal{O} \times \mathcal{O} \mapsto \mathbb{R}$, and $s \in \mathcal{S}$ the current sequence of conformers, we define:

$$r(s, a) = \begin{cases} 0 & \text{if exists } i, \ s.t. \ d(s[i], \mathcal{F}(a)) \leq m, \\ \bar{r}(a) & \text{otherwise} \end{cases}$$

## 2.2 TorsionNet

The TorsionNet model consists of a graph network for node embeddings, a memory unit, and fully connected action layers. TorsionNet takes as input a global memory state and the graph of the current molecule state post-minimization, with which it outputs actions for each individual torsion.

**Node Embeddings.** To extract node embeddings, we utilize a Graph Neural Network variant, namely the edge-network MPNN of Fey and Lenssen [9], Gilmer et al. [12]. Node embedding generates an $M$-dimensional embedding vector $\{\boldsymbol{x}_i\}_{i=1}^N$ for each of the $N$ nodes of a molecule graph by the following iteration:

$$\boldsymbol{x}_i^{t+1} = \boldsymbol{\Theta}\boldsymbol{x}_i^t + \sum_{j \in \mathcal{N}(i)} h\left(\boldsymbol{x}_j^t, \boldsymbol{e}_{i,j}\right),$$

where $\boldsymbol{x}_i^1$ is the initial embedding that encodes location and atom type information, $\mathcal{N}(i)$ represents the set of all nodes connected to node $i$, $\boldsymbol{e}_{i,j} \in \mathbb{R}^D$ represents the edge features between node $i$ and $j$, $\Theta \in \mathbb{R}^{M \times M}$ is a Gated Recurrent Unit (GRU) and $h \in \mathbb{R}^M \times \mathbb{R}^D \to \mathbb{R}^M$ is a trained neural net, modelled by a Multiple Layer Perception (MLP).

**Pooling & Memory Unit.** After all message passing steps, we have output node embeddings $\boldsymbol{x}_i$ for each atom in a molecule. The set-to-set graph pooling operator [12, 47] takes an input all the embeddings and creates a graph representation $\boldsymbol{y}$. We use $\boldsymbol{y}_t$ to denote the graph representation at time step $t$. Up to time $t$, we have a sequence of representations $\{\boldsymbol{y}_1, \ldots, \boldsymbol{y}_t\}$. An LSTM is then applied to incorporate histories and generate the global representation, which we denote as $\boldsymbol{g}_t$.

**Torsion Action Outputs.** As previously noted, the action space $\mathcal{A} \subset \mathcal{O}$ is the torsional space, with each torsion angle chosen independently. The model receives a list of valid torsions $T_j$ for the given molecule for $j = 1, \ldots n$. A torsion is defined by an ordinal succession of four connected atoms as such $T_i = \{b_1, b_2, b_3, b_4\}$ with each $b_i$ representing an atom. Flexible ring torsions are defined differently, but are outside of the scope of this paper. For each torsion angle $T_i$, we use a trained neural network $m_f$, which takes input of the four embeddings and the representation $\boldsymbol{g}_t$ to generate a distribution over 6 buckets: $f_{T_i} = m_f(\boldsymbol{x}_{b_1}, \boldsymbol{x}_{b_2}, \boldsymbol{x}_{b_3}, \boldsymbol{x}_{b_4}, \boldsymbol{g}_t)$. And finally, torsion angles are sampled independently and are concatenated to produce the final output action at time $t$: $\boldsymbol{a}_t = (a_{T_0}, a_{T_1}, \ldots a_{T_n})$, for $a_{T_i} \sim f_{T_i}$.

**Proximal Policy Optimization (PPO).** We train our model with PPO, a policy gradient method with proximal trust regions adapted from TRPO (Trust Region Policy Optimization) [35]. PPO has been shown to have theoretical guarantee and good empirical performance in a variety of problems [22, 36, 55]. We combine PPO with an entropy-based exploration strategy, which maximizes the cumulative rewards by executing $\pi$: $\sum_{t=1}^H \mathbb{E}\left[r_t + \alpha H(\pi(\cdot \mid s_t))\right]$.

**Doubling Curricula.** Empirically, we find that training directly on a large molecule is sampling inefficient and hard to generalize. We utilize a doubling curriculum strategy to aid generalization and sample efficiency. Let $\mathcal{X}_J = \{x_1, \ldots x_J\}$ be the set of $J$ target molecules from some molecule class. Let $\mathcal{X}_J^{1:n}$ be the first $n$ elements in the set.

Our doubling curriculum trains on set $\mathcal{X}_t = \mathcal{X}_J^{1:2^{t-1}}$, by randomly sampling a molecule $x$ from $\mathcal{X}_t$ as the context on round $t$. The end of a round is marked by the achievement of desired performance. The design of doubling curriculum is to balance learning and forgetting as we always have a $1/2$ probability to sample molecules in the earlier rounds (see Algorithm 1 in the appendix).

## 3 Evaluation

In this section, we outline our experimental setup and results[1]. Further details such as the contents of the graph data structure, hyperparameters, and MD experimental setup are presented in Appendix C. To demonstrate the effectiveness of sequential conformer search, we compare performance first to the state-of-the-art conformer generation algorithm RDKit on a family of small molecules, and secondly to molecular dynamics methods on the large-scale biopolymer lignin. All test molecules are shown in Appendix C, along with normalizing constants.

### 3.1 Environment Setup

All conformer search environments are set up using the OpenAI Gym framework [5] and use RDKit for the detection and rotation of independent torsion angles. We use a modular deep RL framework [42] for training. For these experiments, we utilize the classical force field MMFF94, both for energy function evaluation and minimization. The minimization process uses an L-BFGS optimizer, as implemented by RDKit. $Z_0$ and $E_0$ are required for per molecule reward normalization, and are collected by benchmarking on one run of a classical conformer generation method. For the non-stationary reward function described in Section 2.1, we use the distance metric known as the Torsion Fingerprint Deviation [38] to compare newly generated conformers to previously seen ones. To benchmark on nonsequential generation methods, we sort output conformers by increasing energy and apply the Gibbs score function.

Table 1: Method comparison of both score and speed on two branched alkane benchmark molecules. All methods sample exactly 200 conformers. Standard errors produced over 10 runs.

| Method | 11 torsion alkane | | 22 torsion alkane | |
|---|---|---|---|---|
| | Gibbs Score | Wall Time (s) | Gibbs Score | Wall Time (s) |
| RDKit | $1.14 \pm 0.16$ | $11.41 \pm 0.11$ | $1.22 \pm 0.43$ | $68.72 \pm 0.08$ |
| Confab | $0.10 \pm 0.01$ | $\mathbf{10.25 \pm 0.02}$ | $\leq 10^{-4}$ | $\mathbf{26.04 \pm 0.12}$ |
| TorsionNet | $\mathbf{2.38 \pm 0.25}$ | $15.69 \pm 0.03$ | $\mathbf{4.48 \pm 1.86}$ | $35.23 \pm 0.06$ |

## 3.2 Branched Alkane Environment

We created a script to randomly generate molecular graphs of branched alkanes via a simple iterative process of adding carbon atoms to a molecular graph. 1057 alkanes containing $rbn = 10$ are chosen for the train set. The curriculum order is given by increasing number of atoms $|b|$. The validation environment consists of a single 10 torsion alkane unseen at train time. All molecules use sampling horizon $K = 200$. The input data structure of a branched alkane consists of only one type of atom embedded in 3D space, single bonds, and a list of torsions, lending this environment simplicity and clarity for proof of concept. Hydrogen atoms are included in the energy modelling but left implicit in the graph passed to the model. We collect a normalizing $Z_0$ and $E_0$ for each molecule using the ETKDG algorithm, with $E_0$ being the smallest conformer energy encountered, and $Z_0$ being the Gibbs score of the output set with $\tau = 504K$. Starting conformers for alkane environments are sampled from RDKit, and the distance threshold $m$ is set to 0.05 TFD.

**Results.** Table 1 shows very good performance on two separate randomly chosen test molecules, which are 11 and 22 torsion branched alkane examples. Not only does TorsionNet outperform RDKit by 108% in the small molecule regime, but also it generalizes to molecules well outside the training distribution and beats RDKit by 267% on the 22-torsion alkane. TorsionNet's runtime is comparable to Confab's on both trials.

## 3.3 Lignin Environment

We adapted a method to generate instances of the biopolymer family of lignins [31]. Lignin polymers were generated with the simplest consisting of two monomer unit [2-lignin] and the most complex corresponding to eight units [8-lignin]. With each additional monomer, the number of possible structural formulas grows exponentially. The training set consists only of 12 lignin polymers up to 7 units large, ordered by number of monomers for the curriculum. The validation and test molecules are each unique 8-lignins. The Gibbs score reward for the lignin environment features high variance across several orders of magnitude, even at very high temperatures ($\tau = 2000K$), which is not ideal for deep RL. To stabilize training, we utilize the log Gibbs Score as reward, which is simply the natural log of the underlying reward function as such: $r_{log}(s_t, a_t) = \log(\sum_{\tau=1}^{t} r(s_\tau, a_\tau)) - \log(\sum_{\tau=1}^{t-1} r(s_\tau, a_\tau))$. This reward function is a numerically stable, monotonically increasing function of the Gibbs score. Initial conformers for lignin environments are sampled from OpenBabel, and the distance threshold $m$ is set to 0.15 TFD.

## 3.4 Performance on Lignin Conformer Generation

We compare the lignin conformers generated from TorsionNet with those generated from MD. The test lignin molecule has 56 torsion angles and is comprised of 8 bonded monomeric units. RDKit's ETKDG method failed to produce conformers for this test molecule. Since exploration in conventional MD can be slowed down and hindered by high energy barriers between configurations, enhanced sampling methods such as SGMD [6, 51] that speed up these slow conformational changes are used instead. SGMD is used as a more exhaustive benchmark for TorsionNet performance. Structures from the 50 ns MD simulation were selected at regular intervals and energetically minimized with MMFF94. These conformers were further pruned in terms of pairwise TFD and relative energy cutoffs to eliminate redundant and high-energy conformers.

TorsionNet outperforms SGMD in terms of conformer impact toward Gibbs Score (Table 2). Conventional MD is left out from the results, as it only produced 5 conformers that are within pruning cutoffs, mainly due to low diversity according to TFD. This means that exploration was indeed hampered by high energy barriers preventing the trajectory from traversing low energy regions of conformational space. SGMD showed better ability to overcome energy barriers and was able to produce a high

Table 2: Summary of conformer generation on lignin molecule with eight monomers using TorsionNet and molecular dynamics. Standard errors over 10 runs.

| Method | No. of sampled conformers | CPU Time (h) | Gibbs Score |
|---|---|---|---|
| Enhanced MD (SGMD)[1] | 10000 | 277.59 | 1.00 |
| Confab | 1000 | **0.24 ± 0.01** | $\leq 10^{-4}$ |
| TorsionNet[2] | 1000 | 0.35 ± 0.01 | **2.19 ± 1.01** |

[1] Enhanced MD run only once due to computational expense.
[2] All methods are run on CPU at test time to achieve fair comparisons.

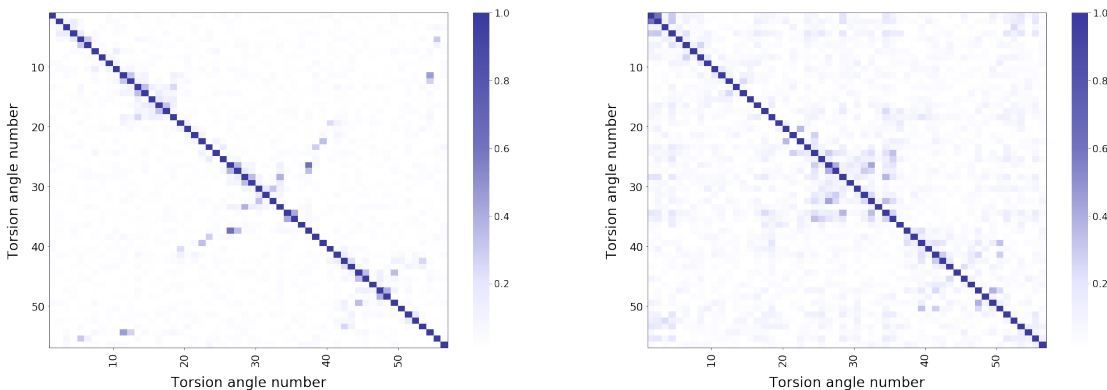

Figure 2: *(best viewed in color)* Torsion angle correlation matrix from SGMD (left) and TorsionNet (right) using lignin's heavy atom torsion angles. Absolute contributions larger than 0.01 are shown. Periodicity of torsion angles is accounted for using the maximal gap shift approach [44].

number of conformers. Although TorsionNet sampled 10x fewer conformers than SGMD, it produces a Gibbs score on average 119% higher, which demonstrates that TorsionNet sampled low-energy unique conformers far more frequently than SGMD. In terms of number of calls to the MMFF energy function, TorsionNet runs 700,000 train time evaluations on non-test lignins and 1000 at test time to achieve the score presented in the paper. To compare, SGMD takes 25 million CHARMM evaluations at 2 fs steps on test lignin. TorsionNet is therefore highly efficient at conformer sampling and captured around twice as much Gibbs score as SGMD at a thousandth of the compute time.

Figure 2 shows the correlated motion of lignin's torsion angles in SGMD and TorsionNet and gives insight toward the preferred motion of the molecule. The highest contributions in SGMD are mostly localized and found along the diagonal, middle, lower right sections of the matrix. These sections correspond to strong relationships of proximate torsion angles, which SGMD identifies as the main regions that induce systematic conformational changes in the lignin molecule. With TorsionNet, we can see high correlations in similar sections, especially the middle and lower right parts of the matrix. This means that TorsionNet preferred to manipulate torsions in regions that SGMD also deemed to be conformationally significant. This result demonstrates that TorsionNet and SGMD behave similarly when it comes to detecting important torsion relationships in novel test molecules.

## 4 On the Benefit of Curriculum Learning

Previous work [3, 49] explains the benefits of curriculum learning in terms of non-convex optimization, while many RL papers point out that curriculum learning eases the difficulties of exploration [10, 28]. Here we show that a good curriculum allows simple exploration strategies to achieve near-optimal sample complexity under a task relatedness assumption involving a joint policy class over all tasks.

**Joint function class.** We are given a finite set of episodic and deterministic MDPs $\mathcal{T} = \{M_1, \dots, M_T\}$. Suppose each $M_t$ has a unique optimal policy $\pi_t^*$. Let $\boldsymbol{\pi}$ denote a joint policy and we use $\boldsymbol{\pi}^*$ if all the policies are the optimal policies. For any set $v \subseteq [T]$ of subscripts, let $\boldsymbol{\pi}_v = (\boldsymbol{\pi}_{v_1}, \dots, \boldsymbol{\pi}_{v_{|v|}})$.

We assume that $\boldsymbol{\pi}^*$ is from a joint policy space $\Pi$. The relatedness of the MDPs is characterized by some structure on the joint policy space. Our learning process is similar to the well-known process of eliminating hypotheses from a hypothesis class as in version space algorithms. For any set $v \in [T]$,

once we decide that $M_{v_1}, \ldots, M_{v_{|v|}}$ have policies $\boldsymbol{\pi}_v$, the eliminated hypothesis space is denoted by $\Pi(\boldsymbol{\pi}_v) = \{\boldsymbol{\pi}' \in \Pi : \boldsymbol{\pi}' = \boldsymbol{\pi}_v\}$. Finally, for any joint space $\Pi'$, we use the subscript $t$ to denote the $t$-th marginal space of $\Pi'$, i.e. $\Pi'_t := \{\pi_t : \exists \boldsymbol{\pi} \in \Pi', \boldsymbol{\pi}_t = \pi_t\}$.

**Curriculum learning.** We define a curriculum $\tau$ to be a permutation of $[T]$. A CL process can be seen as searching a sequence of spaces: $\{\Pi_{\tau_1}, \Pi_{\tau_2}(\hat{\boldsymbol{\pi}}_{\tau_{:2}}), \ldots, \Pi_{\tau_T}(\hat{\boldsymbol{\pi}}_{\tau_{:T}})\}$, where $\tau_{:t}$ for $t > 1$ is the first $t-1$ elements of the sequence $\tau$ and $\hat{\boldsymbol{\pi}}$ is a sequence of estimated policies. To be specific, on round $t = 1$, our CL algorithm learns MDP $M_{\tau_t}$ by randomly sampling policies from marginal space $\Pi_{\tau_1}$ until all the policies in the space are evaluated and the best policy in the space is found, which is denoted by $\hat{\boldsymbol{\pi}}_{\tau_1}$. On rounds $t > 1$, space $\Pi_{\tau_t}(\hat{\boldsymbol{\pi}}_{\tau_{:t}})$ is randomly sampled and the best policy is $\hat{\boldsymbol{\pi}}_{\tau_t}$.

**Theorem 1.** *With probability at least $1 - \delta$, the above procedure guarantees that $\pi^*_{\tau_t} \in \Pi_{\tau_t}(\hat{\boldsymbol{\pi}}_{\tau_{:t}})$ for all $t > 1$ and it takes $O(\sum_{t=1}^{T} K_{\tau_t} |\Pi_{\tau_t}(\boldsymbol{\pi}^*_{\tau_{:t}})| \log^2(T|\Pi_{\tau_t}(\boldsymbol{\pi}^*_{\tau_{:t}})|/\delta))$ steps to end.*

The proof of Theorem 1 is in Appendix A. In some cases (e.g. combination lock problem [20]), we can show that $\sum_{t=1}^{T} K_{\tau_t} |\Pi_{\tau_t}(\boldsymbol{\pi}^*_{\tau_{:t}})|$ matches the lower bound of sample complexity of any algorithm. We further verify the benefits of curriculum learning strategy in two concrete case studies, combination lock problem (discussed in Appendix B) and our conformer generation problem.

## 4.1 Conformer generation

**Problem setup.** We simplify the conformer generation problem by finding the *best* conformers (instead of a set) of $T$ molecules, where it becomes a set of bandit problems, as our stationary reward function and transition dynamic only depend on actions. We consider a family of molecules, called T-Branched Alkanes (see Appendix C) satisfying that the $t$-th molecule has $t$ independent torsion angles and is a subgraph of molecule $t + 1$ for all $t \in [1, T]$.

**Joint policy space.** The policy space $\Pi_t$ is essentially the action space $\Pi_t = \mathcal{A}_0^t$, where $\mathcal{A}_0 = \{k\pi/3\}_{k=1}^{6}$. Let $a_t^*$ be the optimal action of bandit $t$. We make Assumption 2 for the conditional marginal policy spaces of general molecule families.

**Assumption 2.** *For any $t \in [T]$, $a_t^* \in \Pi_t(a_{t-1}^*) := \{a \in \Pi_t : d_H(a_{1:t-1}, a_{t-1}^*) \leq \phi(t)\}$, where $d_H(a_t^1, a_t^2) := \sum_{i=1}^{t} \mathbf{1}(a_{ti}^1 \neq a_{ti}^2)$ for $a_t^1, a_t^2 \in \mathcal{A}_t$ is the Hamming distance. Note that in our T-Branched Alkanes, $\phi(t) \approx 0$.*

**Sample complexity.** Applying Theorem 1, each marginal space is $\Pi_t(a_{t-1}^*)$ and the total sample complexity following the curriculum is upper bounded by $\tilde{O}(\sum_{t=1}^{T} |\mathcal{A}_0|^{\phi(t)+1})$ with high probability and learning each molecule separately may require up to $\sum_{t=1}^{T} |\mathcal{A}_0|^t$, which is essentially larger than the first upper bound when $\phi(t) < t - 1$. When $\phi(t)$ remains 0, the upper bound reduces to $T|\mathcal{A}_0|$.

**Effects of direct parameter-transfer.** While it is shown above that a purely random exploration within marginal spaces can significantly reduce the sample complexity, the marginal spaces are unknown in most cases as $\phi(t)$ is an unknown parameter. Instead, we use a direct parameter-transfer and entropy based exploration. We train TorsionNet on 10 molecules of T-Branched Alkanes sequentially and evaluate the performances on all the molecules at the end of each stage. As shown in Figure 3, the performance on the hardest task increases linearly as the curriculum proceeds.

## 5 Conclusion and Outlook

Posing conformer search as an RL problem, we introduced the TorsionNet architecture and its related training platform and environment. We find that TorsionNet reliably outperforms the best freely available conformer sampling methods, sometimes by many orders of magnitude. We also investigate the results of an enhanced molecular dynamics simulation and find that TorsionNet has actually uncovered more of the conformational space than seen via the more intensive sampling method. These results demonstrate the promise of TorsionNet and DeepRL methods in conformer generation of large-scale high *rbn* molecules. Such methods open up the avenue to efficient conformer generation on any large molecules without conformational databanks to learn from, and to solve downstream tasks such as mechanistic analysis of reaction pathways. Furthermore, the curriculum-based RL approach to combinatorial problems of increasing complexity is a powerful framework that can extend to many domains, such as circuit design with a growing number of circuit elements, or robotic control bodies with increasing levels of joint detail.

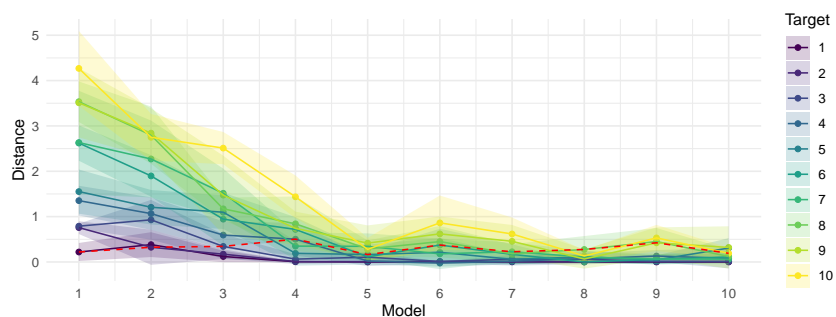

Figure 3: We train a set of models sequentially on molecules indexed by $\{1, 2, \ldots, 10\}$ from the T-Branched Alkanes. Axis $x$ represents the model trained on molecule $x$ with parameters transferred from model $x - 1$. Axis $y$ represents the distance in energy between the conformation predicted by model $x$ and the best conformer for target $y$ marked by the colors. The confidence interval is the one standard error among 5 runs. Red dashed line marks the one-step transferring performance.

Our work is a first step toward solving the conformer generation problem using deep reinforcement learning. There are many opportunities for further work. First, the vast chemical space beyond lignin and branched alkanes is worth exploring. Second, some molecules may have rotationally-equivalent conformers, for example, conformations with methyl groups, which may undercount the free energy of symmetrical configurations. Future work can work on extensions to deal with such symmetry issues. Finally, to generate test molecules, we used simple incremental generation for branched alkanes and fragments of lignin. Future work can consider more sophisticated molecular generation methods [40, 52].

## 6    Broader Impacts

The reported viability of TorsionNet signifies that it can be applied to conformer generation of relevant large flexible molecules in other areas such as chemistry and materials science. The investigation on the non-fossil carbon source lignin helps inform targeted depolymerization strategies to yield valuable products for applications such as renewable energy.

**Acknowledgements**

We acknowledge the support of NSF via grants CAREER IIS-1452099 and CHE-1551994.

## Footnotes

[1]Our code is available at `https://github.com/tarungog/torsionnet_paper_version`.

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
