[Supplementary Material]

# A  Proof of Theorem 1

Recall the Theorem 1.

*The optimal policy $\pi_{\tau_t}^*$ is guaranteed in $\Pi_{\tau_t}(\hat{\boldsymbol{\pi}}_{\tau:t})$ for all $t \geq 1$. With probability at least $1 - \delta$, the algorithm takes at most $O(\sum_{t=1}^T K_{\tau_t} |\Pi_{\tau_t}(\boldsymbol{\pi}_{\tau:t}^*)| \log^2(T|\Pi_{\tau_t}(\boldsymbol{\pi}_{\tau:t}^*)|/\delta))$ steps to end. A curriculum-free algorithm that learns tasks separately requires samples at least $\sum_{t=1}^T K_t |\Pi_t|$.*

For the first argument, we use induction. On round $t$, assuming

$$\pi_{\tau_t}^* \in \Pi_{\tau_t}(\hat{\boldsymbol{\pi}}_{\tau:t}), \tag{1}$$

we have $\hat{\boldsymbol{\pi}}_t = \pi_{\tau_t}^*$. Then for $t + 1$, equation (1) also holds. As $\pi_{\tau_1}^* \in \Pi_{\tau_1}$, the argument follows by induction. For the second part, we it is essentially a Coupon Collector's problem.

**Lemma 3** (Coupon Collector's problem). *It takes $O(N \log^2(N/\delta))$ rounds of random sampling to see all $N$ distinct options with a probability at least $1 - \delta$.*

*Proof.* Consider a general sampling problem: for any finite set $\mathcal{N}$ with $|\mathcal{N}| = N$. For any $n$, whose sampling probability is $p(c)$, with a probability at least $1 - \delta$, it requires at most

$$\frac{\log(1/\delta)}{\log(1 + \frac{p(n)}{1-p(n)})} \text{ for } n \text{ to be sampled.}$$

Since $\log(1 + x) \geq x - \frac{1}{2}x^2$ for all $x > 0$, we have

$$\frac{\log(1/\delta)}{\log(1 + \frac{p(n)}{1-p(n)})} \leq \log(1/\delta) \frac{1}{\frac{p(n)}{1-p(n)} - \frac{p(n)^2}{2(1-p(n))^2}} = O(\log(1/\delta) \frac{1-p(n)}{p(n)}).$$

Searching the whole space $\mathcal{N}$ with each new element being found with probability $\frac{N-i}{N}$ at round $i$, it requires at most

$$O(\sum_{i=1}^N \log(\frac{N}{\delta}) \frac{N}{N-i}) = O(\log^2(\frac{N}{\delta})N),$$

with a probability at most $1 - \delta$.

By Lemma 3, with a probability $1 - \delta/T$, search the marginal policy space $\Pi_{\tau_t}(\boldsymbol{\pi}_{:\tau_t})$ requires at most $O(K_{\tau_t} \log^2(T|\Pi_{\tau_t}(\boldsymbol{\pi}_{:\tau_t})|/\delta)|\Pi_{\tau_t}(\boldsymbol{\pi}_{:\tau_t})|)$ times policy evaluation. As the horizon for task $\tau_t$ is $K_t$, the total number of samples to search the whole joint space is

$$\sum_{t=1}^T K_{\tau_t} |\Pi_{\tau_t}(\boldsymbol{\pi}_{:\tau_t})| \log^2(T|\Pi_{\tau_t}(\boldsymbol{\pi}_{:\tau_t})|/\delta).$$

# B  Combination lock

**Problem setup.** We consider the combination lock problem [20]. As shown in Figure 4, the set of $T$ MDPs $\{M_1, \ldots, M_T\}$ share the same action space $\mathcal{A} = \{-1, +1\}$. The $t$-th task has the state space $\mathcal{S}_t = \{1, \ldots, t\}$, the episode length $t$. The agent receives 0 reward on all but the last state $t$ in the $t$-th task. There are two actions, one for staying on the current state and the other one for moving forward, i.e. $s_{t+1} = s_t + 1$.

Figure 4: Combination lock MDPs.

**Joint policy space.** We assume the same optimal actions on the common states shared by different tasks. Formally, $\pi_{t_1}(s, h_1) = \pi_{t_2}(s, h_2)$ for $t_2 \geq t_1$, $s \in \mathcal{S}_{t_1}$ and $h_1 \in [t_1], h_2 \in [t_2]$.

Figure 5: We trained a set of models sequentially on gridworld problem with size $\{3, 5, \ldots, 17\}$. Model $x$ is the model trained on environment $x$ using the parameters transferred from model $x - 1$. The colors represent the target environment. Each point $(x, y)$ in the plot represents the distance in rewards between the conformer suggested by model $x$ and the optimal reward. The red dashed line links the points of test environments $x + 1$ using the model trained on environment $x$. The confidence interval is based on the standard deviation over 100 episodes.

**Sample complexity.** By [50], the total number of steps needed to learn $M_T$ is at least $AT^3$. The lower bound can only be achieved by carefully designed exploration strategy, which accounts for the underlying function class. Applying Theorem 1, a purely random exploration strategy following curricula $M_1, \ldots, M_T$ has an upper bound of $O(\sum_{t=1}^{T} H_t |\Pi_t(\pmb{\pi}_t)| \log(\frac{\sum_{t=1}^{T} |\Pi_t(\pmb{\pi}_t)|}{\delta})) = \tilde{O}(AT^3)$ with probability at least $1 - \delta$, which matches the lower bound. Solving $M_T$ directly using random exploration requires $O(2^T)$ samples.

**Experiment setup.** To match the experiment setup in our conformer generation problem, we conduct the combination lock experiment on a harder environment, MiniGrid. MiniGrid is a minimalistic gridworld environment for OpenAI Gym with an image input. The environment is shown in Figure 6. In our experiments, we train an PPO on MiniGrid of size 25, with target grid changing according to the sequence $\{(3, 3), (5, 5), (7, 7), \ldots, (17, 17)\}$. The model setting and hyper-parameters are the same in Torch-rl. Whenever the model converges on the current task, we test the average regret over 100 samples on all the tasks from 3 to 17. The results are shown in Figure 5. As we can see, we observe a similar pattern as shown in Figure 3.

Figure 6: MiniGrid environment of size 6: an agent takes actions from {Turn Left, Turn Right, Move Forward} to reach the target grid (green). The starting grid is always placed in the left-up corner (1, 1) of the gridworld. A positive reward 1 is received only when the agnet reaches the target grid.

# C Algorithm Details and Experimental Parameters

## C.1 Curriculum Algorithm

---
**Algorithm 1** TorsionNet trained with doubling curriculum

---
Initialize model parameter $\theta$, round $t = 1$, the sequence of target molecule $\mathcal{X}_J$, starting set $\mathcal{X}_1 = \{\mathcal{X}_J[1]\}$;
**for** round $t = 1, \ldots, T$ **do**
    **while** True **do**
        1. Sample a molecule $x$ from $\mathcal{X}_t$
        2. Train on $x$ with TorsionNet.
        **if** Performance Threshold Reached **then**
            3. Set $\mathcal{X}_{t+1} \leftarrow \mathcal{X}_t$
            4. Add molecules from $\mathcal{X}_J$ to $\mathcal{X}_{t+1}$ until $|\mathcal{X}_{t+1}| = 2|\mathcal{X}_t|$
            5. $\mathcal{X}_J \leftarrow \mathcal{X}_J \backslash \mathcal{X}_{t+1}$
            6. Break
        **end if**
    **end while**
**end for**

---

The specifics of our implementation are included with the code.

## C.2 Features and Hyperparameters

Table 3: Molecule Features

| Feature | Feature Type | Description | Dimensionality |
|---|---|---|---|
| Atom type | Node | [C, O] (one-hot) | 2 |
| Position | Node | 3D Cartesian coordinates (float) | 3 |
| Bond type | Edge | [Single, Double, Triple, Aromatic] (one-hot) | 4 |
| Conjugated | Edge | Bond belongs to a conjugated system (boolean) | 1 |
| Ringed | Edge | Bond is in a closed ring (boolean) | 1 |

Position of atoms are given by Cartesian coordinates. These are taken directly from the RDKit conformer object, then normalized in two ways. Firstly atoms are centered on the origin. Then, rotation is normalized such that eigenvectors align with coordinate axes.

Table 4: Experimental Constants

| Molecules | $E_0$ (kcal/mol) | $Z_0$ | $\tau(^\circ K)$ |
|---|---|---|---|
| 11-torsion alkane | 7.840935037731404 | 13.066560104213275 | 503 |
| 22-torsion alkane | 14.882782943326 | 1.2363186365185 | 503 |
| 8-lignin | 525.8597422 | 16.1548792743065 | 2000 |

$E_0$ and $Z_0$ are utilized for Gibbs evaluation. Normalizers for alkane train and test molecules are sampled from RDKit ETKDG with default settings, and for the lignin test environment via exhaustive SGMD sampling. The lignin train molecules have normalizers collected via OpenBabel sampling. We include the constants for test molecules here, but all remaining constants for train molecules are included in code repository in Appendix E.

Table 5: Selected Hyperparameters

| Hyperparameter | Value |
|---|---|
| Message Passing Steps | 6 |
| Set-to-Set Passes | 6 |
| Node Embedding Dimension | 128 |
| LSTM Hidden State Dimension | 256 |

Full hyperparameter setup described in code repo (Appendix E).

## C.3 Test Molecule Depiction

(a) 11-torsion alkane

(b) 22-torsion alkane

Figure 7: Stick visualization of alkane test molecules with implicit hydrogen atoms. (black: carbon)

Figure 8: Stick visualization of 8-lignin molecule with implicit hydrogen atoms. (black: carbon, red: oxygen)

(a) T-Alkane 0         (b) T-Alkane 4         (c) T-Alkane 9

Figure 9: Stick visualization of T-Branched Alkane molecule family with implicit hydrogen atoms. Each subsequent T-alkane is a superset of the molecular graph of the prior T-alkane, with one additional carbon on the long end. (black: carbon)

Full smiles string is given for each molecule in code repo (Appendix E).

### C.4 Molecular Dynamics Computational Details

The lignin oligomer topology was obtained using Lignin-KMC [31] and 3D coordinates were generated with OpenBabel's gen3D [29] and optimized with molecular mechanics. CHARMM [6] was the software used for the molecular dynamics simulations. Parametrization of the system was done with the CHARMM General Forcefield (CGenFF) [46]. The simulations were carried out with Langevin dynamics in vacuum at 300K with a collision frequency of 10 per ps. The nonbonded list cutoff was set at 14 angstroms and interactions were modulated by a switching function between 10 and 12 angstroms. The shake constraint was used to fix bond lengths involving hydrogen atoms. The simulations involved 2 ns of heating and 50 ns of production at 2 fs timestep. The self-guided dynamics settings involved a local average time of 0.2 ps and momentum guiding factor of 1. The coordinates in the production run were saved every 5 ps for subsequent analysis.

## D  Diversity of conformer sets

We calculate the RMSD (root-mean-square deviation) of every pair of conformers of 8-Lignin generated by SGMD and TorsionNet. The former has 2352 pruned conformers and the latter has 986. As shown in Figure 10, both methods have similar distribution for the pair-wises RMSDs with a range roughly in $[4, 10]$ angstroms.

Figure 10: Histograms of pairwise RMSDs of two conformers sets, one from SGMD (left) and the other one from TorsionNet (right). The unit of distance for the $x$-axis is angstrom.

## E  Code

Github link: `https://github.com/tarungog/torsionnet_paper_version`