[Reviews · NeurIPS 2020]

Review 1

Summary and Contributions: The submission presents a reinforcement learning based sequential conformer search approach for generating low energy 3D geometries of molecules. The model is trained by curriculum learning and is benchmarked against a few commonly used sequential conformer search models. Additionally, a novel evaluation metric is proposed, based on the Gibbs distribution of generated conformers.

Strengths: An interesting problem with potentially important applications in materials science and drug discovery

Weaknesses: Evaluation setup. The test datasets are extremely small (only a handful of molecules), and I have some concerns with the proposed Gibbs metric Some implementation details are missing, eg the MD simulation setup No code is provided

Correctness: I do not think the current experimental setup is sufficient to draw strong conclusions. In particular, the results are reported for only 3 individual test molecule examples (a 11 torsion alkane, 22 torsion alkane, and a 8-lignin biopolymer). From my understanding of the newly proposed Gibbs metric, I have some concerns, especially since it is the key metric that is used in the experiments. The normalization constants (Z0, E0) for the two alkane examples are from the lowest energy conformer obtained from the ETKDG method. It seems to me that you would want the normalization constants to be calculated from as accurate a method as possible (since it is in some ways assumed to be a ground truth of some sort), but is the ETKDG method actually expected to be more accurate than Confab or TorsionNet?

Clarity: Some parts of the paper could be made clearer, such as section 3

Relation to Prior Work: The related work section references two broad review papers about the conformer generation literature, however it could be a bit more explicit with a few specific prior works and how they relate to this submission. Eg "Bayesian optimization for conformer generation" https://jcheminf.biomedcentral.com/articles/10.1186/s13321-019-0354-7

Reproducibility: No

Additional Feedback: The Gibbs metric seems to conflate the quality of the conformers (in terms of energy) with the diversity of the conformers, and by itself may not give a complete picture of the conformer set that is generated. Perhaps it would be informative to also include some individual metrics about the minimum energy of the conformer set, and about the diversity of the set. Update after rebuttal: Author addressed some of my concerns in rebuttal


Review 2

Summary and Contributions: This paper applies reinforcement learning to generate diverse low-energy conformations of molecules. It essentially accelerates brute-force exhaustive approaches that sampel every possible combination of dihedral angle orientation in rigid rotors

Strengths: This paper has a number of excellent new ideas and contributions that can find wide application in computational chemistry, even if they are not broadly applicable to machine learning. Conformer generation is far from a solved problem in molecular simulation and current tools do a good job, but they are either cheap and not powerful or extremely powerful and extremely expensive. Anything that’s faster and/more effective is a big win. Reinforcement learning is an excellent choice of tool for this problem: the action and state space can be clearly stated in terms of rigid body rotations. The proposed metric is intuitive and very relevant. The baselines are relatively strong and the proposed model outperforms them.

Weaknesses: Because the idea is new and very interesting, a number of topics can up that could/should be addressed. Is there a way to be certain that the gradient descent using MMFF has the molecule stay on the same basin of the PES that the rigid rotor sampled? It is likely, particularly in crowded conformations that the structure and energy that MMFF reports are not for the same internal angles as the initial torsion angles would suggest. The Gibbs Score is introduced as some completely new idea, but it’s essentially related to a (relative) population according to Maxwell Boltzmann statistics. Furthermore, the log of Gibbs score is then a relative free energy, a very intuitive connection with the underlying physics. The Gibbs score is naturally biased by the thermal accessibility of conformations, so for potential energy surfaces with deep wells it will return just 1 one for the global minimum and nothing else. I wonder what the implications are for this conformer generation task. In drug binding to protein, the binding free energy for a given conformer might be with a relatively high energy conformer that represents say 0.01% of the equilibrium conformational ensemble but 100% of the bound pose, because of great selectively. In this case, al algorithm capable of accessing low-population states is important. How was de-duplication applied to the ETKDG results from RDKit? The stochastic sampling from the distance matrix approach can (and will) produce multiple exact copies of the same conformer, with the same energy, which may artificially boost their Gibbs score. How does TorsionNet take into account symmetry and its effect of free energy? There are multiple ways to re-label atoms and keep the same exact connectivity and 3D structure, and each of this is a state in terms of free energy. For instance, for every conformation with methyl, there are 3 rotationally-equivalent orientation for the methyl group. This is taken into account by other tools (https://xtb-docs.readthedocs.io/en/latest/crest.html) The rejection approach would undercount the free energy of symmetrical configurations. Confab is exhaustive, so there is very little reason to expect it will find 200 meaningful conformers in the first 200 attempts. It would be a nice benchmark for the T alkanes to also show the full outcome of confab (which should also give an upper ceiling for the Gibbs score) “motion” is Line 249 is a slightly confusing word choice. TorsionNet is a static, recursive approach that does not really have motion (as in time-evolution) in the way molecular dynamics does. How many calls to the MMFF energy function were needed at train time in aggregate? How does this compare with inference with CONFNET and with the baselines? How is the transferability across chemical space? The alkanes are the same bond over and over, and in the case of lignin the train and test case are the same. The MPNN embeddings will struggle in unseen bond combinations. What is the Gibbs score of the non-enhanced MD? Even if it found few conformers, if they are very low energy the could represent most of the phase space. Does Confab run FF minimization after each rotation? Otherwise it should be much faster than TorsionNet to sample the same number of conformers. Where does the first starting conformation come from? Lastly – how does random search over rotations look like? Confab does a poor job at few attempts because it is exhaustive, MD takes a long time to jump over barriers, and TorsionNet needs lots of evaluation at train time (this should be reported as a succinct metric) How much would one get from running a Monte Carlo search over dihedrals? (equivalent to torsionnet with a uniform distribution as a policy)

Correctness: “Molecular dynamics (MD) simulations (though expensive) are the usual method for sampling complex molecules such as lignin [31, 44], but this is a brute force approach that is applied due to a lack of other good options” MD is not brute force, exhaustive enumeration is brute force, MD by definition samples low-energy states more. In particularly advanced sampling approaches like metadynamics are definitely not brute force and they are used very often for conformer generation. The RDKit and OpenBabel baselines are two notable _open source_ methods. Corina is a commonly used commercial too. Lignin is not _one_ polymer, it is a class of polymers. Just choosing one lignin oligomer is not very representative of a mixture of the natural polymer, which is typically cross-linked and over 70 monomers long.

Clarity: Excellent writing, easy to follow and great balance with the technical detail. Some paragraphs are missing line numbering.

Relation to Prior Work: Excellent differentiation from protein folding, which is a related, buy distinct task.

Reproducibility: Yes

Additional Feedback: The author feedback has addressed most of the criticisms. Authors are encouraged to share and open-source their. Some lingering challenges about generalization remain, in particular to unseen chemical bonds and functionalities, and to long-range interactions that GCN's may not address well. Those are however, on the data-acquision and representation learning side, and this RL approach should be transferable as more data and better embedding functions are made available.


Review 3

Summary and Contributions: The authors introduce TorsionNet, a sequential conformer search technique based on reinforcement learning using rigid rotors as an approximation. They show that TorsionNet outperforms other chemoinformatics methods by up to 4x. Although, it is not completely clear from the text the existence of other really competing methods. In general, the advantages of TorsionNet over RDKit and Confab is very clear, showing the benefits of the well thought dedicated NN architecture.

Strengths: The authors introduce a concrete well thought solution to the problem which shows very clear advantages over other methods (at least on the presented examples).

Weaknesses: To my view, more experiments should be done to truly give statistical robustness to the method. The code should have been provided.

Correctness: From the methodological point of view, the theory behind looks strong and the numerical results are good, but a code to personally run at least a toy example would have been more informative and conclusive.

Clarity: The article is very clear, presenting a nice statement of the problem that a general reader can enjoy. The technical part is in parts dry (due to the complexity of the system) but still understandable.

Relation to Prior Work: This is nicely presented.

Reproducibility: No

Additional Feedback: In section 3.2, the authors should briefly mention (or in the introduction) other methodologies to generate molecular graphs. For example MCTS as used in https://www.nature.com/articles/s42005-020-0338-y or http://papers.nips.cc/paper/8974-symmetry-adapted-generation-of-3d-point-sets-for-the-targeted-discovery-of-molecules


Review 4

Summary and Contributions: The authors implement a novel RL algorithm to sample diverse conformations of small molecules.

Strengths: The researchers proposed a novel framework for sampling molecular conformations with RL + deep learning. I think the utilization of previously defined energy functions to be of increased interest in the future. I also think that implementing these methods in numerically stable ways is very challenging, so I applaud the authors for actually making it work!Generally, though I agree that the authors’ Wall Time evaluations in Table 1 and Table 2 are correct, I do not see this as a particularly strong argument. Confab and RDkit are used because they are easily accessible, not particularly because they are fast. This work is concerned about novel methods for sampling, not the implementation. The authors’ code is presumably using a GPU for TorsionNet, and presumably MCMC can be sped up with GPUs. If Wall Time is an evaluation metric, it is only fair if those algorithms are using the same hardware. For other comments, please see the correctness section.

Weaknesses: Generally, though I agree that the authors’ Wall Time evaluations in Table 1 and Table 2 are correct, I do not see this as a particularly strong argument. Confab and RDkit are used because they are easily accessible, not particularly because they are fast. This work is concerned about novel methods for sampling, not the implementation. The authors’ code is presumably using a GPU for TorsionNet, and presumably MCMC can be sped up with GPUs. If Wall Time is an evaluation metric, it is only fair if those algorithms are using the same hardware. For other comments, please see the correctness section.

Correctness: I’m not convinced by the argument to use the Gibbs score. Since it is a new statistic, it needs to be compared to previous statistics to show its validity and usefulness. How does the score compare to an all-by-all distance matrix of conformations? In what situations does the Gibbs Score do better than an all-by-all distance calculation? Without that shown, I’m not convinced it is any better. I would like to see this as an additional column in Table 1 and Table 2, or at least in the Supplement. It would also be great to compare conformations to a ground truth dataset. Are there any datasets that exist that have validated diverse structures? In particular, this is necessary for the much-discussed lignin example: Are the conformations realistic? As a trained chemist, are there actually any examples of how MD simulations of lignin are useful for energy production? The two papers cited are purely computational, and the link to actual energy production is lacking.

Clarity: Generally, I think the work does an okay job of laying out what you are working towards in the paper. I would like a bit more description of MMFF. It is unclear to me what is being shown in the curriculum learning experiments. What does the y axis on Figure 3 represent? How does curriculum learning work while training your model? Finally, this paper still operates under the small molecule regime. As noted in the “Prior Work” section, many researchers are interested in conformations of proteins, which are much larger than lignin researched here, and other realistic molecules presented in this work.

Relation to Prior Work: I fundamentally, and very strongly disagree, that the protein folding problem is not working on molecular dynamics, as the most certainly are. Entire research organizations are working on this: https://www.deshawresearch.com/publications.html. An example of an ML approach for sampling from an ensemble is here: Hernández, C.X., Wayment-Steele, H.K., Sultan, M.M., Husic, B.E. and Pande, V.S., 2018. Variational encoding of complex dynamics. Physical Review E, 97(6), p.062412. Please also cite other work that uses energy-based sampling and generative models as well, and please compare against: Ingraham, J., Riesselman, A.J., Sander, C. and Marks, D.S., 2019. Learning Protein Structure with a Differentiable Simulator. In ICLR. David Belanger and Andrew McCallum. Structured prediction energy networks. In International Conference on Machine Learning, pp. 983–992, 2016. Though RDkit and OpenBabel are the easiest for the authors to implement, they are most likely not the fastest. What about other MC algorithms, like replica exchange, or any method presented here: Yang, Y.I., Shao, Q., Zhang, J., Yang, L. and Gao, Y.Q., 2019. Enhanced sampling in molecular dynamics. The Journal of chemical physics, 151(7), p.070902.

Reproducibility: Yes

Additional Feedback:

[Author Response · NeurIPS 2020]

We thank the reviewers for their very helpful remarks. Overall, we accepted all small corrections, cited the additional literature, and added missing details pointed out by the reviewers. In the following, we address common concerns and then individual reviewers' questions, but to our regret, space limits dictate that we cannot respond to every point. Multiple reviewers pointed out the lack of attached code, for which we apologize. Our code is not proprietary, and we plan to fully release it with the camera-ready version. Unfortunately attachments are not allowed for rebuttals. Reviewers also commented that our test data is a small set. From a chemistry perspective, the tested lignin molecule with multiple linkages is already a complex system that involves many viable conformers, wherein good sampling performance can provide chemically rich information toward studying depolymerization reaction pathways. Even this single large lignin molecule required *weeks of compute time* to gather enhanced MD results. In fact, it is this very compute cost that our method attempts to address. Moreover, note that TorsionNet *generalizes over many molecules during the training curriculum*. Nonetheless, we are adding more alkane eval environment data for the final submission.

**Response to Reviewer 1.** 1. *Concerns on Gibbs score.* The normalization constants ($Z_0$, $E_0$) are *not* assumed to be "ground truth". Normalization is used for (1) numerical stability of rewards and (2) interpretability of results. Furthermore, Gibbs free energy is always **relative** and force field methods mostly concern themselves with energy *differences* rather than absolutes. As Reviewer 2 comments, this metric is intuitive and grounded in the physics.

**Response to Reviewer 2.** First, we graciously accept Reviewer 2's positive remarks and thank them for their support. 1. *Connection of Gibbs score to other statistics/ bias to thermal accessibility.* We thank Reviewer 2 for pointing out that (log) Gibbs score is deeply grounded in physics and related to the Boltzmann distribution as relative population and relative free energy. We present it as "new" as it has not been used in conformer generation literature, and as a "score" to be clear to an ML audience. In general, drug screening moves in the direction of finding more accessible conformers; good bindings must have relatively low Gibbs free energy. Furthermore, while the Gibbs score is designed as a generic metric, we can bias the reward for specific tasks and continue using the developed methods.

2. *De-duplication of ETKDG/symmetry concerns.* The output of all compared methods (incl. ETKDG) goes through a minimization step, and then a distance exclusion function to remove all near duplicates. Similarly, while Gibbs may be undercounted due to symmetries, the comparison is fair since we apply the same scoring uniformly. We agree the symmetry issue is important, and must be investigated when dealing with highly symmetrical molecules (lignin is not).

3. *Concerns about Confab/random search over rotations.* To clarify a misconception: Confab essentially **is** a random search over rotations. It iterates in a random order over all possible conformers (O'Boyle et al., 2011). Therefore, sampling the first $N$ of this exhaustive search is similar to TorsionNet with uniform distribution policy, except without replacement. Running it to completion on the T-alkanes experiment is not necessary, as in this case we are not looking for the entire partition function, but merely the lowest energy conformation (which can be identified by eye).

4. *Number of calls to MMFF.* It takes around 500,000 evaluations on **non-test** lignins and 1000 at inference time to achieve the score presented in the paper. To compare, SGMD takes 25 million Charmm evals at 2 fs steps on test lignin.

**Response to Reviewer 3.** We thank Reviewer 3 for their positive remarks.

**Response to Reviewer 4.** 1. *Unfair CPUtime/walltime comparison.* TorsionNet is not using GPU speedup at inference time. *We specifically ran it on CPU to achieve fairer test results*. We will add more experiment detail to appendix.

2. *Gibbs score vs. all-by-all distance.* It is not clear to us what metric should be used to compare metrics, and we consider our Gibbs score an attempt to import the well established thermodynamics principle of free energy into the context of conformer generation as observed by Reviewer 2. Nonetheless, we agree that analyzing the diversity of our generated conformers via all to all distance in the supplement would be valuable, and will do so.

3. *Ground truth for lignin.* There are only a handful of experimental crystal structures that have been published (Vermaas et al., 2019) and they are mostly limited to small dimeric structures only. Data from MD serve as the ground truth dataset, comprised of physics-based conformers, similarly to (Simm and Hernández-Lobato, 2019).

4. *Are MD simulations of lignin useful for energy production and any paper that talks about it?* Scientists employing mechanochemical experimental processes (where MD's ability to model many conformers is important) to extract renewable energy from plant biomass argue that: "Understanding wood and lignin processing on a molecular level appears essential for improving their degradation efficiencies." (Kleine et al., 2013)

5. *Why CL experiments? How is CL used?* The CL experiments are simple models to demonstrate the validity of our theoretical work. Fig 3. y-axis shows the distance in energy between the best sampled conformer of the current model and the global best conformer. We observe how errors drop for unseen test molecules as more examples are added to the train set. We train TorsionNet sequentially from small molecules to large as described in Secs. 3.2,3.3 and App. C.1.

6. *Protein folding IS working on MD.* We agree, and are unaware of anything in our paper that suggests otherwise. Protein folding commonly uses enhanced sampling MD methods (we do as well in our benchmarks).

[Meta-Review · NeurIPS 2020]

The reviewers found this paper to be interesting and compelling, nicely summarized by R2 in discussion: think the method is sound and exciting and the key challenges in transferability live in the availability of (high-accuracy) training data and in the challenges of representation learning for molecules (GCNs need to be exposed to a lot of chemical variability to be able to interpolate in chemical space.). The alkanes are essentially the same bond over and over and lignin is trained and tested in the same chemical space. I insist that these are representation learning challenges to be solved by the community and improvements there could be combined with this RL approach." That said, the reviewers did find several areas where the paper can be improved. Because of space limitations, I understand that not all of these suggestions will be able to be incorporated within page limits, but I do expect the authors will address as much as possible within the main final text, and all feedback addressed either in main text or in a supplementary appendix.